# Direct Numerical Simulation of Surface Wrinkling for Extraction of Thin Metal Film Material Properties

**DOI:** 10.3390/mi14040747

**Published:** 2023-03-28

**Authors:** Seonho Seok, HyungDal Park, Philippe Coste, Jinseok Kim

**Affiliations:** 1Center for Nanoscience and Nanotechnology (C2N), University-Paris-Saclay, 91400 Orsay, France; 2Center for Bionics, Korea Institute of Science and Technology (KIST), Seongbuk-gu, Seoul 02792, Republic of Korea

**Keywords:** direct simulation, FEM, surface wrinkling, materials properties

## Abstract

This paper presents a direct numerical simulation for the extraction of material properties based on thin-film wrinkling on scotch tape. Conventional FEM-based buckling simulation sometimes requires complex modeling techniques concerning mesh element manipulation or boundary conditions. The direct numerical simulation differs from FEM (finite element method)-based conventional two-step linear–nonlinear buckling simulation in that mechanical imperfections are directly applied into the elements of the simulation model. Hence, it can be performed in one step to find the wrinkling wavelength and amplitude, which are key parameters to extract the material mechanical properties. Moreover, the direct simulation can reduce simulation time and modeling complexity. Using the direct model, the effect of the number of imperfections on wrinkling characteristics was first studied, and then wrinkling wavelengths depending on the elastic moduli of the associated materials were prepared for the extraction of material properties. Thin-film wrinkling test patterns on scotch tape were fabricated using the transfer technique with low adhesion between metal films and the polyimide substrate. The material properties of the thin metal films were determined by comparing the measured wrinkling wavelengths and the proposed direct simulation results. By consequence, the elastic moduli of 300 nm thick gold film and 300 nm thick aluminum were determined as 250 GPa and 300 GPa, respectively.

## 1. Introduction

Surface wrinkling driven by mechanical instability is commonly observed in thin-film structures with a compliant substrate, and it has recently been proposed as a tool to find the material properties of thin-film materials [1,2,3,4,5]. In general, the material properties of a thin film deposited on a substrate can be determined on the basis of the measured curvature caused by the residual stress of a thin film using the Stoney equation [6,7,8]. The substrate curvature can be measured using different methods, namely, mechanical methods, capacitance methods, X-ray diffraction methods, and optical methods. Most of these techniques provide a measure of the out-of-plane detection of the curved film–substrate system due to the film residual stress. Other interesting techniques are direct measurements of material properties such as nanoindentation and nanoscratch tests [9,10]. The strip bending test is performed by applying an external load to a micromachined fixed-fixed strip specimen with a nanoindenter and analyzing the resulting load-displacement characteristics. The strip bending test can provide the tensile properties of thin films such as elastic modulus and stress [11].

As an alternative method, surface wrinkling based on the thin-film buckling phenomenon, mentioned earlier, has recently been proposed to determine the elastic moduli of thin films such as thin polymer films, thin film metals, and carbon nanotubes [12,13,14]. This technique allows calculating the elastic modulus of a thin film by measuring the wavelength of the wrinkling thin film. Elastomer materials such as PDMS are frequently used as a base substrate because of their low elastic modulus. Commercial scotch tape having adhesive on a supporting membrane such as PVC, PET, or PE has also been reported as a buckling substrate with an elastic thin film [15].

For most reports on wrinkling studies, the analytical formulation was widely employed in analyzing experimental data, e.g., wrinkling wavelengths [16]. As for finite element modeling associated with instabilities, two steps of pre-buckling and post-buckling analysis of commercial software packages such as ABAQUS and ANSYS have frequently been used in studies on the surface wrinkling phenomenon [17,18,19,20]. Geometrical imperfection methods have frequently been employed using mesh, geometry, and boundary condition perturbation techniques for studying surface instability. However, the existing imperfection approaches are relatively complex, and the implementation can be laborious due to free parameter calibration. Furthermore, the interpretation and verification of results are not straightforward, which makes it less practical for common users. To overcome the drawbacks of geometrical imperfection methods, a simulation method which is direct, robust, and relatively easy to perform has been proposed, wherein the pre/post-buckling simulations can be performed successfully in only one analysis step. In addition, it can be implemented with any common finite element code and analysis platform [21,22,23]. For the extraction of material properties, thin metal film wrinkling on commercial scotch tape was reported in our previous work [15] using conventional linear–nonlinear buckling simulations through commercially available ANSYSY software.

In this paper, the extraction of material properties based on scotch tape surface wrinkling and direct numerical simulation is proposed. Direct numerical simulation of the thin-film buckling is a simple and efficient solution to obtain the wrinkling wavelength and amplitude, which are essential parameters for the extraction of material properties. The theoretical background of thin-film wrinkling on compliant substrate is explained in Section 2, and the FEM model for direct numerical simulation is described in Section 3. Section 4 presents the results and discussion of the FEM modeling and simulation. Results from the fabrication process of thin-film wrinkling and the extraction of material properties are presented in Section 5. Lastly, the conclusion is provided in Section 6.

## 2. Theory of Wrinkling of Thin Film for Extraction of Material Properties

The force balance approach for surface buckling instability of a thin film on a compliant substrate is briefly introduced below [1]. Considered as a semi-infinite substrate under plane strain deformation, the classical equation for bending of an elastic film on a compliant elastic substrate is given by Equation (1).
(1)$$\bar{E_{f}}I\frac{d^{4}z}{dx^{4}}+F\frac{d^{2}z}{dx^{2}}+kz=0,$$
where $\bar{E}$ = *E*/(1 − *ν*^2^) is the plane-strain modulus, *E* is the Young’s modulus, *v* is the Poisson’s ratio, *I* = wh^3^/12 is the moment of inertia (where *w* is the width of the film, and *h* is its thickness), *F* is the uniaxially applied force or load, and *k* is Winkler’s modulus of an elastic half-space (*k* = wπ/λ). The subscripts *f* and *s* denote the film and substrate, respectively.

As the buckling instability of interest here is the first sinusoidal mode, the film deflection can be described by Equation (2).
(2)$$z\left(x\right)=A\sin\frac{2\pi x}{\lambda }.$$

Substituting Equation (2) into Equation (1) and solving for the applied force in the thin film on compliant substrate yields Equation (3).
(3)$$F=4\bar{E_{f}}I{\left(\frac{\pi }{\lambda }\right)}^{2}+\frac{\bar{E_{s}}w}{4}{\left(\frac{\pi }{\lambda }\right)}^{-1}.$$

The film buckling wavelength can be found by minimizing *F* with respect to *λ* (or, ∂F/∂λ=0):(4)$$\lambda =2\pi h{\left(\frac{\bar{E_{f}}}{3\bar{E_{s}}}\right)}^{\frac{1}{3}}.$$

It should be noted that the wavelength is only a function of the thickness of the film and the elastic properties of the film and substrate. Thus, the wrinkling wavelength can be used to determine the material properties of the thin film if the substrate material properties are known.

Another important parameter is the critical stress or strain needed to induce the wrinkling in the system. The critical stress can be found from Equations (3) and (4) by dividing critical force (*Fc*) by the cross-sectional area of the thin film:(5)$$\sigma_{c}=\frac{F_{c}}{hw}={\left(\frac{9}{64}\bar{E_{f}}\;{\bar{E_{s}}}^{2}\right)}^{\frac{1}{3}}.$$

Therefore, the critical strain is given as below.
(6)$$\varepsilon_{c}=\frac{\sigma_{c}}{\bar{E_{c}}}=\frac{1}{4}{\left(\frac{3\bar{E_{S}}}{\bar{E_{f}}}\;\right)}^{\frac{2}{3}}.$$

Under the assumption that the wrinkling wavelength is independent of the applied strain, wrinkling amplitude can be found using Equation (7).
(7)$$A=h\sqrt{\frac{\varepsilon -\varepsilon_{c}}{\varepsilon_{c}}},$$
where *A* is the wrinkling amplitude, and *ε* is the applied strain.

## 3. FEM Model Description for Direct Numerical Simulation

The direct FEM simulation for thin-film wrinkling on compliant substrate was carried out using a 2D film–substrate system, as shown in Figure 1. The surface wrinkles were simulated using the plane strain model. Note that the plane strain assumption is consistent with the theoretical formulation in Equations (5)–(7). The dimensions of the 2D model were defined with a width of 250 µm, depth of 50 µm, and thin-film thickness of 100 nm.

According to the direct numerical simulation, a 2D model was built with varied imperfections embedded in compliant substrate [21,22,23]. These embedded imperfections make it possible to conduct a buckling simulation without using the conventional linear–nonlinear buckling simulation procedure. The boundary conditions of Ux = 0, Uy = free at x = 0, Ux = −1 µm, Uy = free at x = L, and Ux = free, Uy = 0 at y = 0 were applied as shown below. Note that the symmetry plane was applied at x = 0. The applied displacement of Ux was adjusted to tune the applied strain, thereby changing the wrinkling amplitude. The material properties needed for the modeling were the Young’s modulus and Poisson ratio: 6.9 MPa and 0.45 for compliant substrate; 50 GPa and 0.42 for thin film. The mechanical properties of the compliant substrate were experimentally measured in our previous work [15]. Wrinkling of the thin film was confirmed, as shown in Figure 2.

## 4. Direction Simulation Results and Discussions

As the imperfection defined in the substrate is the essential element of the direct simulation, the effect of the spacing between the imperfections on wrinkling behavior was checked in terms of wrinkling wavelength and amplitude, as shown in Figure 3. Interestingly, the wrinkling wavelength did not depend on the spacing, whereas the wrinkling amplitude showed a high degree of dependence. It can be said that the wrinkling amplitude proportionally increased with the spacing between the imperfections. Thus, it was studied in more detail by comparing the wrinkling patterns resulting from the variant spacings between the imperfections. The wrinkling patterns were sinusoidal, as shown in Figure 4a, because the first buckling mode was dominant in the case of thin-film wrinkling on compliant substrate. Envelopes of the sinusoidal wrinkling patterns were extracted as shown in Figure 4b in order to find the uniformity of the wrinkling amplitude. It was found that (1) the minima of the wrinkling amplitude occurred where the imperfections were defined, (2) the highest wrinkling amplitudes were found at the edges, and (3) uniform wrinkling amplitude along the length was achieved with nine imperfections. Note that nine imperfections resulted in 25 µm spacing between imperfections, which was most similar to the wrinkling wavelength of 24 µm (see Figure 3).

For the extraction of material properties, the wrinkling wavelength was measured from the device under test (DUT) explained in the previous work [15]. The thin-film wavelength measured from the test pattern was used to determine the material properties; thus, the thin-film wrinkling wavelength was prepared as a function of thin-film elasticity for various substrate elasticity, as shown in Figure 5. This figure includes the analytical calculation, conventional FEM modeling, and simulation results based on eigenvalue linear-nonlinear buckling and FEM direction simulation results. A comparison of three different approaches revealed that the wrinkling wavelength from the direct simulation was close to the analytical calculation for a lower elastic modulus, whereas it increased somewhat for a larger elasticity. The larger elasticity created a larger bending force in the wrinkling pattern, which was the cause of deviation between the simulation and analytical calculation. It should be noted that the direct simulation was achieved with one imperfection. Referring to the previous measurement of wrinkling wavelength, the elastic modulus of the gold thin film was estimated at 147 GPa by the conventional FEM simulation when the substrate elasticity was assumed to be 6.9 MPa. The direct FEM simulation produced an elastic modulus of 200 GPa for the given wrinkling wavelength and substrate elasticity.

Wrinkling amplitude was used as an indicator of the residual stress of the thin film of interest; thus, it was prepared as a function of applied strain for certain elastic moduli of substrate and film. Before addressing wrinkling amplitude, the critical strain to induce surface wrinkling was studied as a function of the material properties, as well as the elasticities of the thin film and substrate. Figure 6 shows the critical strain as a function of the elasticity of the thin film with different substrate elasticity. The critical strain required for surface wrinkling initiation was inversely proportional to the thin-film elasticity as expected. Furthermore, the substrate elasticity had the same tendency as the thin film in that the critical strain was reduced at lower substrate elasticity.

Next, the wrinkling amplitude was studied using the nine-imperfection model, which resulted in uniform wrinkling amplitude along the model width as mentioned earlier. Figure 7 shows the wrinkling amplitude variation as a function of the applied strain. For comparison, the analytical calculation is also plotted with both the conventional linear–nonlinear simulation and the direct simulation. The disparity between the analytical calculation and the simulations may have been caused by the limited substrate thickness of the simulation models. The analytical equation for critical strain was extracted by assuming an infinite substrate thickness, as explained in the previous section.

## 5. Fabrication of Thin-Film Wrinkling and Extraction of Material Properties

Test samples were fabricated to determine the material properties of thin metal films which were patterned on a polymer substrate, photosensitive polyimide (PSPI, HD4100, HD MicroSystems™, Parlin, NJ, USA), as shown in Figure 8. To make the metal wrinkling patterns on scotch tape, thin metal films were deposited on a PSPI layer built on Si substrate by e-beam evaporation (ei-5k, ULVAC, Chigasaki, Japan). The PSPI layer was used as it has low adhesion force with metal films if an adhesion layer such as titanium or chromium is not introduced. First, the native oxide and organic matter formed on the Si wafer surface were removed through a general piranha cleaning solution (a 3:1 mixture of sulfuric acid and 30% hydrogen peroxide). Thereafter, the PSPI was coated to a thickness of 5 μm with 4500 rpm/min conditions using a spin coater. The coated PSPI was soft-baked at 110 °C for 6 min, and the PSPI was exposed to 200 mJ UV using the aligner equipment (MA-6 Mask Aligner, Karl Suss, Munich, Germany). Afterward, PEB (post-exposure baking) was performed at 110 °C for 5 min, and a pattern was developed using a dedicated developer (401D, HD MicroSystems™, Parlin, USA) and Rince (AZ Thinner 1500, MERCK, Rahway, USA). The patterned PSPI was subjected to PDB (post-development baking) at 180 °C for at least 1 min, and full curing was performed at 200 °C for 30 min at 200 °C and at 300 °C for 1 h in a vacuum drying oven (Customized Oven, SH Scientific Co., Ltd., Sejong, Republic of Korea). Al (aluminum) and Au (gold) were individually deposited on a full-cured PSPI to a thickness of 150 and 300 nm using an E-beam evaporator. Adhesion layers such as Ti and Cr, which are generally used to improve the adhesion force between PSPI and thin metal film, and surface treatment using CF_4_-based plasma did not apply sufficient debonding of thin metal film using scotch tape. Patterning was performed on the deposited metal thin film using a positive photoresist (GXR-601 46cp, AZ Electronics Materials, Hsinchu County, Hsinchu, Taiwan), and Al and Au were selectively dry-etched using an ICP-RIE (inductively coupled plasma reactive ion etcher, Oxford Instruments, Abingdon, UK). Considering the low adherence of the formed Al and Au pattern with PSPI, the photoresist removal was completed using O_2_ plasma at 300 W for 5 min using Asher equipment (Plasma Finish V15-G, Ebhausen, Germany).

The debonding force was measured using commercial tensile experimental equipment (Shimadzu EZ-S machine, Shimadzu, Kyoto, Japan). The maximum applied force was about 1.4 N in average, as shown in Figure 9. Long and narrow thin metal layers were transferred onto the scotch tape using a peel-off process, as described in our previous study [15]. After metal transfer, metal film wrinkling patterns were established on the scotch tape due to a mismatch of mechanical properties between the two materials. Figure 10 shows the metal patterns transferred to the scotch tape.

The wrinkling patterns were characterized using an optical microscope and 3D optical profiler in order to measure wrinkling wavelength and amplitude, and then the results were compared with those from the FEM (finite element method) buckling simulation for the extraction of material properties. Figure 11 shows images of the wrinkled thin film transferred to the scotch tape taken using an optical microscope and 3D profiler. The measured wavelengths of the wrinkling pattern are summarized in Table 1. The disparity of the two measurements mainly stemmed from the uncertainty of wrinkling patterns on the optical microscope images.

The wrinkling wavelengths of the thin films were measured at 34 µm for the 150 µm thick Al film, 55 µm for the 300 µm thick Au film, and 58 µm for the 300 µm thick Al film. The measured wrinkling wavelengths were matched in order to determine the corresponding elastic moduli of the thin films, as shown in Figure 12. As a consequence, the elastic moduli of the 300 nm thick gold film and 300 nm thick aluminum were determined as 250 GPa and 300 Gpa when the elastic modulus of the scotch tape was assumed to be 3.5 MPa. The elastic modulus of the 150 nm thick aluminum was 60 GPa in the same conditions as before. Note that the wavelength of the 150 nm thick Au film could not be measured due to the uncertainty of the wrinkling pattern. The extracted elastic moduli of the thin films were larger than those of the bulk materials, and it is found that the elastic moduli of aluminum thin films showed thickness dependence in the range of several hundred nanometers.

## 6. Conclusions and Perspectives

A direct and easy numerical simulation with common finite element simulators was presented to overcome the drawbacks of the geometrical imperfection methods. In general, finite element simulations associated with buckling instabilities have been achieved with pre-buckling and post-buckling analysis of commercial software packages such as ABAQUS and ANSYS; thus, the modeling and simulation are complex and time-consuming. Furthermore, it is known that geometrical imperfection methods frequently employed in buckling studies depend on mesh, geometry, and boundary condition perturbation techniques, requiring laborious free parameter calibration. Furthermore, the interpretation and verification of results are not straightforward, which makes them less practical for common users. Therefore, direct numerical simulation, simply defining the imperfection of the material at the interface of the two associate materials, was proposed, and it was successfully demonstrated that the wrinkling simulation results could be directly determined from the surface wrinkling simulation results in terms of wrinkling wavelength and amplitude. As a practical application, the proposed direct simulation technique was applied to extract material properties on the basis of commercial scotch-tape surface wrinkling. The established direct FEM model successfully provided wrinkling wavelength as a function of expected material properties such as elastic modulus. For the test wrinkling structure, thin metal film wrinkling was implemented by transferring thin films of gold and aluminum onto scotch tape. The elastic modulus of the transferred thin metal films was determined by comparing the measured wrinkling wavelength with the direct simulation results. In conclusion, the direct geometrical imperfection method is an easy and versatile technique to perform buckling simulations, which can also be applied to other devices based on the buckling phenomenon.

## Figures and Tables

**Figure 1 micromachines-14-00747-f001:**
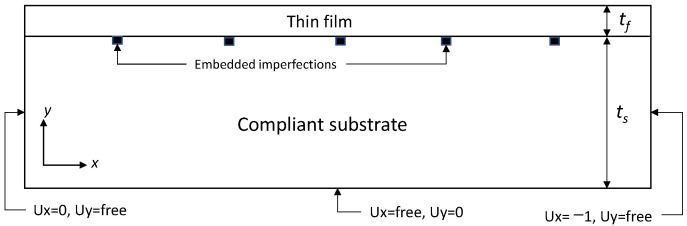
The 2D wrinkling model and boundary conditions.

**Figure 2 micromachines-14-00747-f002:**
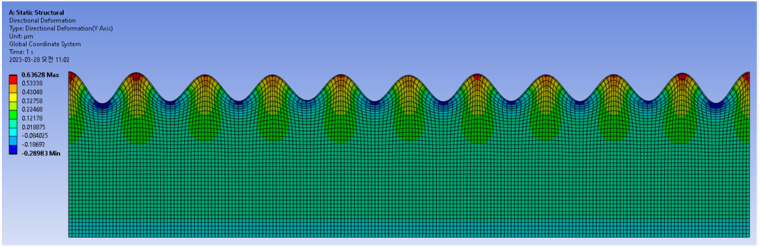
Buckled thin film due to applied strain.

**Figure 3 micromachines-14-00747-f003:**
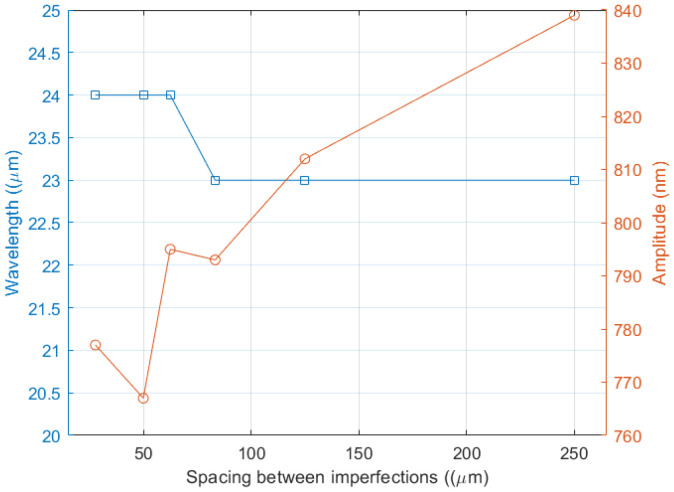
Wrinkling wavelength and amplitude vs. spacing between imperfections.

**Figure 4 micromachines-14-00747-f004:**
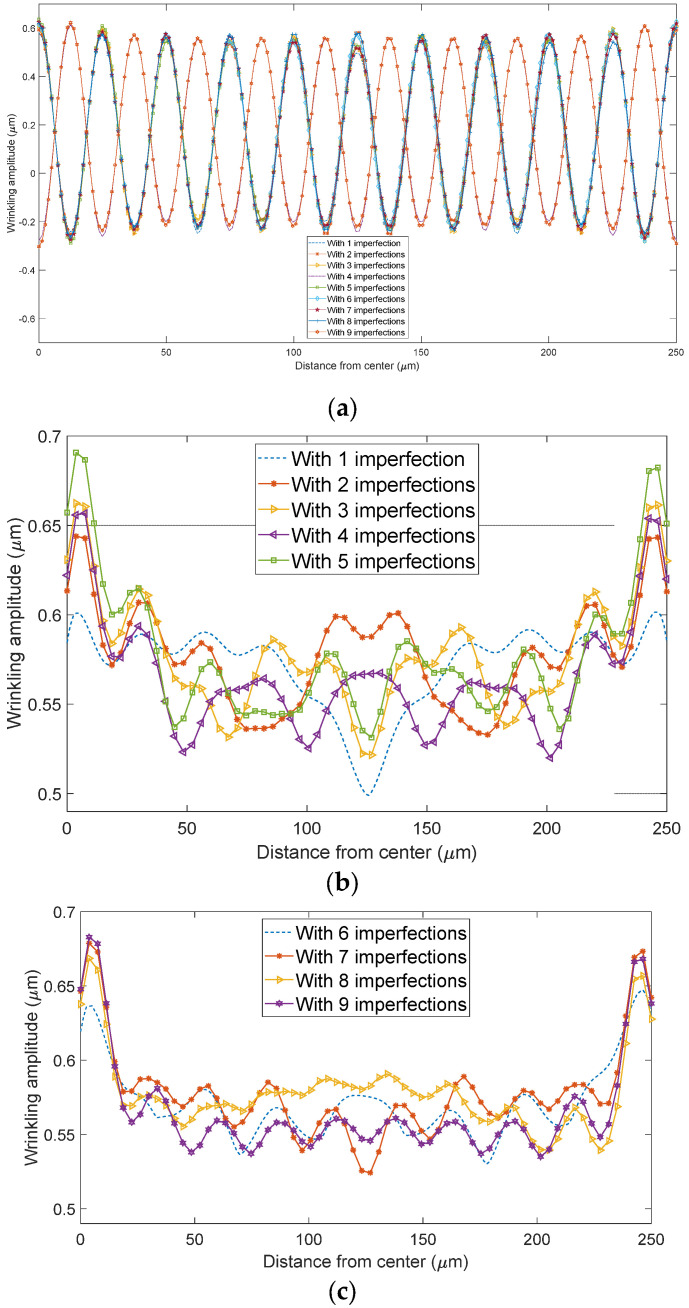
Wrinkling pattern as function of the number of imperfections. (**a**) Sinusoidal wrinkling pattern; (**b**) Envelope of the wrinkling pattern 1; (**c**) Envelope of the wrinkling pattern 2.

**Figure 5 micromachines-14-00747-f005:**
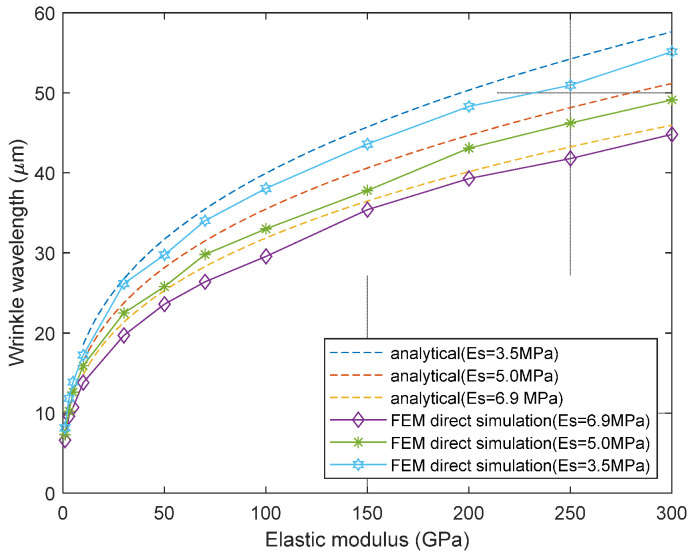
Wrinkling wavelength vs. elastic modulus of thin film as a function of substrate elasticity.

**Figure 6 micromachines-14-00747-f006:**
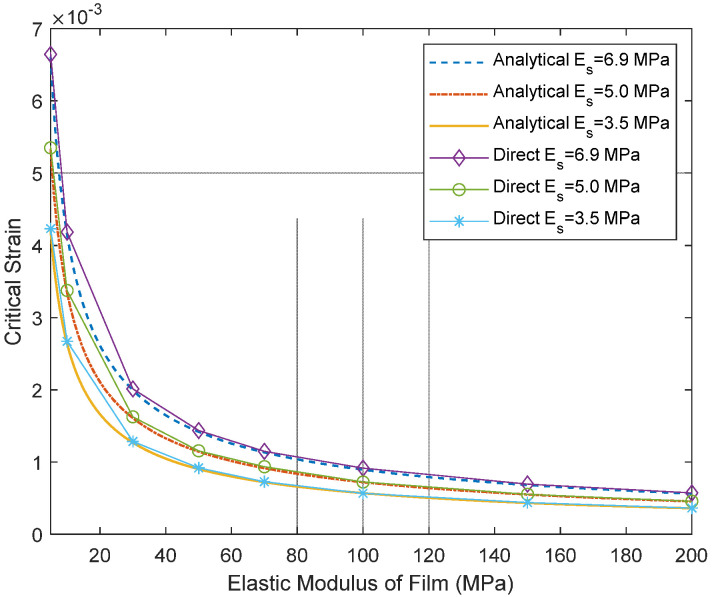
Critical strain as a function of elasticity of thin film for different substrate elasticity.

**Figure 7 micromachines-14-00747-f007:**
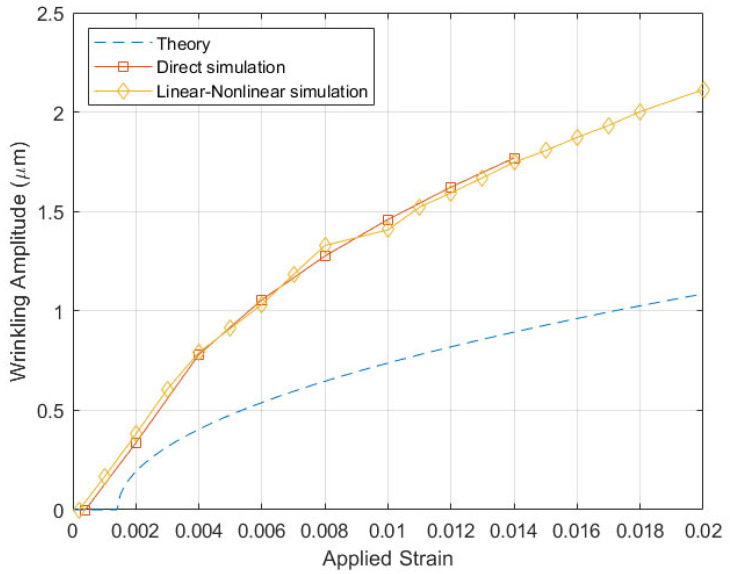
Wrinkling amplitude as a function of applied strain.

**Figure 8 micromachines-14-00747-f008:**
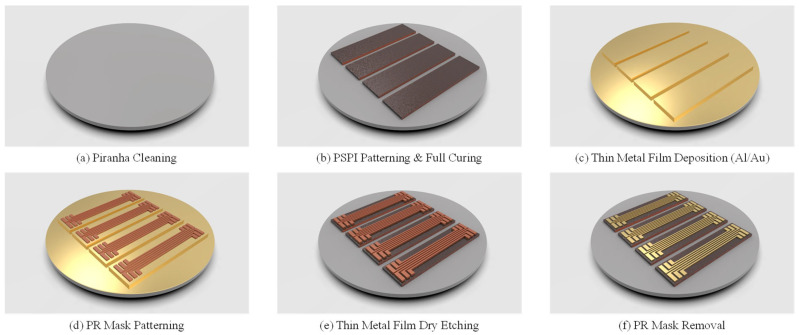
Thin-film metal test pattern fabrication process. (**a**) Piranha cleaning for removing organic residues from Si substrates, (**b**) Spin−coating, UV exposure, developing process, and Full curing in vacuum condition for PSPI patterning, (**c**) Thin metal film (Al and Au) deposition using an E−beam evaporation process. (**d**) Positive photoresist patterning for thin metal film etching mask, (**e**) The ICP−RIE−based dry etching process for thin metal film patterning, (**f**) The Oxygen plasma−based dry etching process for photoresist removing.

**Figure 9 micromachines-14-00747-f009:**
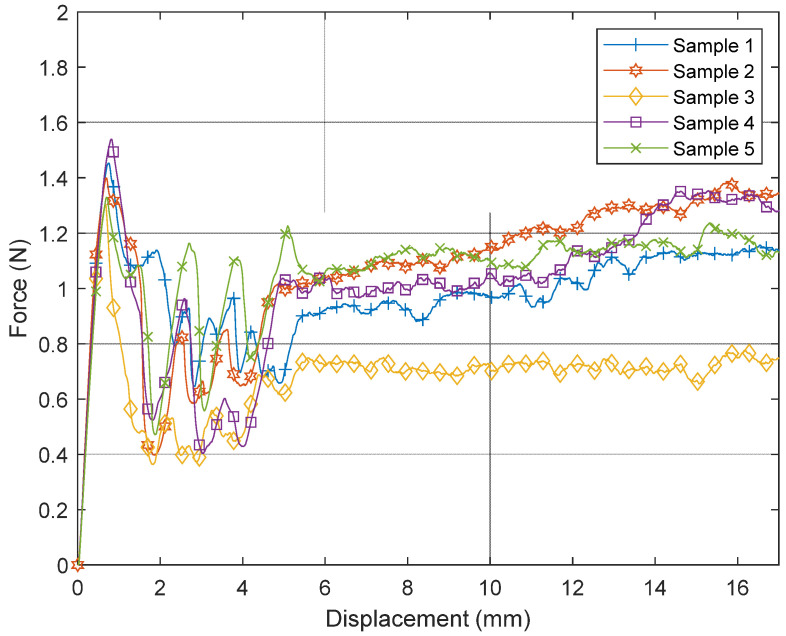
Debonding force measurement of the scotch tape.

**Figure 10 micromachines-14-00747-f010:**
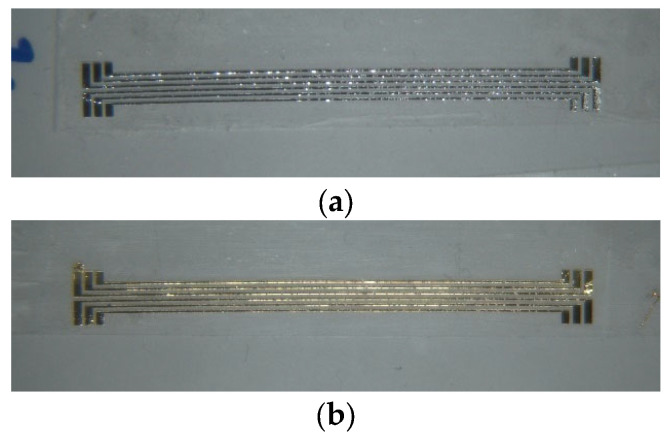
Test structures transferred to scotch tape. (**a**) Al 300 nm; (**b**) Au 300 nm.

**Figure 11 micromachines-14-00747-f011:**
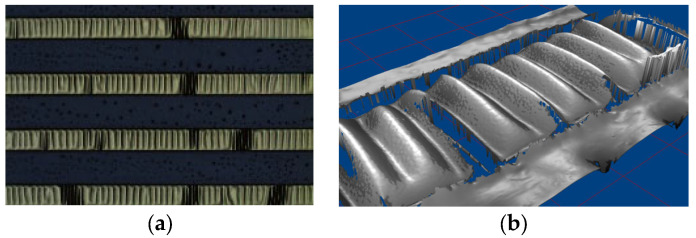
Wrinkling pattern and 3D measurement. (**a**) Optical microscope; (**b**) 3D profiler.

**Figure 12 micromachines-14-00747-f012:**
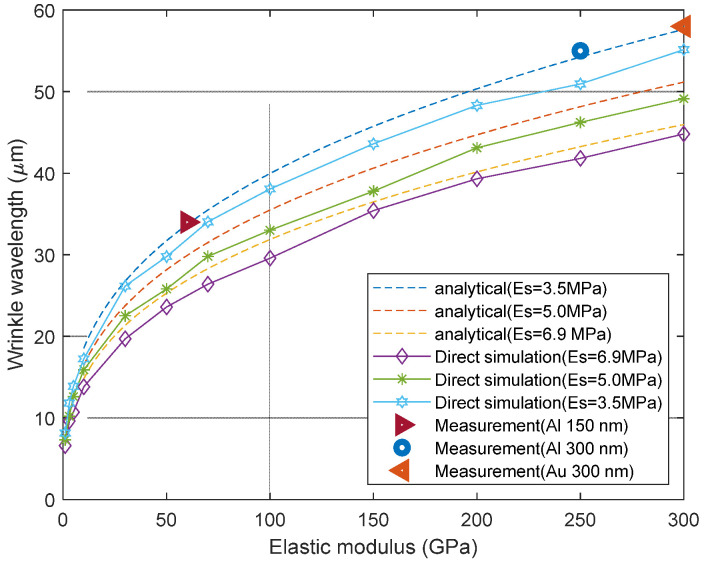
Elastic modulus extracted from wrinkling wavelength.

**Table 1 micromachines-14-00747-t001:** Wavelength of the wrinkling metal pattern.

Thin Films	Wavelength by Optical Microscope	Wavelength by 3D Profiler
Au 150 nm	Non-measurable	Non-measurable
Au 300 nm	55 µm	53 µm
Al 150 nm	34 µm	44 µm
Al 300 nm	58 µm	55 µm

## Data Availability

Not applicable.

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
