# Peer review of "Direct Numerical Simulation of Surface Wrinkling for Extraction of Thin Metal Film Material Properties"

_micromachines, 2023, doi:10.3390/mi14040747_

Round 1

Reviewer 1 Report

The authors proposed the following manuscript: "Direct Numerical Simulation of Surface Wrinkling for Thin Metal Film Material Properties Extraction". The Introduction part as well as section 2-Theory of wrinkling of thin-film for material properties extraction is quite well written. 

At section 3-FEM-model description for direct numerical simulation, the authors need to explain the following: "Compliant substrate has 6.9 MPa and 0.45, while thin film has 50 GPa and 0.42." What is the substrate and the thin film and what is the meaning of those values? If it's Young's modulus and Poisson ration, then correct it.  

In Figure 5, the authors specify three different values for analytical and FEM direct simulation. What do these values represent? In the text, above is a value of 6.9 MPa for the substrate. Why was this value chosen? The same for Figure 6 and 10. 

There is no picture of the final structures. Please correct. 

There is little data on the experimental results extracted after the fabrication of the test structures. Also, the Conclusions part is as good as non-existent.

Author Response

Firstly, I’d like to thank you for the useful comments and suggestions.

The answers to the comments and the revised items have been prepared in the following.

The authors proposed the following manuscript: "Direct Numerical Simulation of Surface Wrinkling for Thin Metal Film Material Properties Extraction". The Introduction part as well as section 2-Theory of wrinkling of thin-film for material properties extraction is quite well written. 

At section 3-FEM-model description for direct numerical simulation, the authors need to explain the following: "Compliant substrate has 6.9 MPa and 0.45, while thin film has 50 GPa and 0.42." What is the substrate and the thin film and what is the meaning of those values? If it's Young's modulus and Poisson ration, then correct it. 

  1. A) The values represent Young’s modulus and poisson ratio, respectively. The text has been revised correctly. 

In Figure 5, the authors specify three different values for analytical and FEM direct simulation. What do these values represent? In the text, above is a value of 6.9 MPa for the substrate. Why was this value chosen? The same for Figure 6 and 10. 

  1. A) The Young’s modulus has been measured 6.9 MPa by tensile test in our previous work and thus this value has been used as default value.

There is no picture of the final structures. Please correct. 

  1. A) Fabricated test structures have been added in Fig. 8.

There is little data on the experimental results extracted after the fabrication of the test structures. Also, the Conclusions part is as good as non-existent.

  1. A) The measurement data useful for this work is wrinkling wavelength. It is compared with the simulation results as presented. Conclusion has been revised.

Reviewer 2 Report

The manuscript entitled “Direct Numerical Simulation of Surface Wrinkling for Thin Metal Film Material Properties Extraction”, authored by Seonho Seok, HyungDal Park, Philippe Coste and Jinseok Kim has been reviewed. 

This paper describes a direct and easy numerical simulation technique based on buckling of thin films to understand the thin-film wrinkling on scotch tapes. “The effect of imperfections on wrinkling characteristics and wrinkling wavelength are depended on the elastic moduli of the associated materials” is the hypothesis used for extracting material properties.  The authors propose a direct simulation method which is relatively easy to perform in a single analysis step to overcome the drawbacks of the geometrical imperfection methods. It is very interesting to note that the established direct FEM model has successfully provided wrinkling wavelength as function of material properties associated with surface wrinkling. In this work, the thin metal film wrinkling has been implemented by transferring thin films of gold and aluminium onto a scotch tape and then the elastic modulus of the transferred thin metal films has been found by comparing the measured wrinkling wavelength with the direct simulation results.

Although the manuscript is very well written, there are few important corrections regarding equation numbers, definition of modelling parameters in the equations are either missing or displayed wrong in the manuscript. The English grammar and sentence structure of the manuscript need to be edited. The overall manuscript needs to be revised thoroughly before publishing it. The scientific content of the paper is novel, very interesting and adequate for publication after making the following corrections.

Q1. P1, L 8-9, “This paper presents direct numerical simulation for thin-film wrinkling on scotch tape for material properties extraction based on buckling of thin film”.

Pease correct the sentence.

Q2. P1, L 9-10, “The direct numerical simulation differs from FEM-based conventional linear-nonlinear buckling simulation”

Please elaborate “FEM” in the above sentence.

Q3. P1, L 25-27, “Most of these techniques provide a measure of the out-of-plane detection of the curved film-substrate system due to the film residual stress”.

What causes the residual stress on thin films? How can we avoid/decrease its effect?

Q4. P1, L 28-29, Like all other mechanical measurements, is there any ASTM standards we should follow while conducting the Strip bending test? Are there any size and finishing limit that we should follow while machining the fixed-fixed strip specimen for successful evaluation of the mechanical properties?

Q5. P1, L 36-39, “Elastomer material such as PDMS is frequently used as a base substrate as it has low elastic modulus and commercial scotch tape having adhesive on a supporting membrane such as PVC, PET, PE was also reported as a buckling substrate with elastic thin film”.

Please Correct the English Grammar of the sentence

Q6. P2, L 13-16, “Section IV shows the results and discussions of the FEM modeling and simulation and thin film wrinkling fabrication process and material properties extraction results will be presented in Section V. Finally, conclusion will be made in section VI.

Please Correct the sentence structure, spelling and English Grammar of the sentence

Q7. P2, L-23, “, where  is the plane-strain modulus”

According to equation (1) ,   ? Also on Line 23, Please remove “,”

Q8. P2, L 31-33, “Substituting Equation (2) into Equation (1) and solving for the applied force in the thin film on compliant substrate gives Equation (3).

There is no Equation (3) listed in the manuscript. Please change the applied force (P2, L33) equation number as Equation (3). Also please correct all the following equation numbers, both on the equations as well as on the text accordingly.

Q9. P3, L1, “wrinkling amplitude can be found by Equation (8)”.

Wrinkling amplitude

Please mark the appropriate equation number to the wrinkling amplitude, describe it in the text as well. Also please define εc

Q10. P3, L 4-5, “The direct FEM simulation for thin film wrinkling on compliant substrate has been carried out through 2D film-substrate system as shown in Fig. .”.

Please indicate the Figure number in the text.

Q11. P3, L8, “The dimension of the 2D model is defined with width of 250 um, depth of 50 um”.

Please indicate the micrometers as “µm” in the manuscript.

Q12. P3, L12-13, “The boundary conditions of Ux=0, Uy=free at x=0 and Ux=-1 um, Uy=free at x=L and Ux=free, Uy=0 at y=0 have been applied as shown below”.

Please indicate the micrometers as “µm”

Q13. P3, L 27-29, “It can be said that the wrinkling amplitude is proportionally increased with the spacing. Thus, it is studied in more detail by comparing the wrinkling patterns depending on the spacing between the imperfections”.

Please correct the English Grammar.

Q14. P5, L 5-7, “The thin-film wavelength measured from the test pattern is the key parameter to determine the material properties and thus the relationship between thin-film wrinkling wavelength and its elastic modulus as function of substrate elasticity is prepared as shown in Fig. 5”.

Please correct the English Grammar.

Q15. P5, L 9-12, “Comparison among three different approaches reveals that the wrinkling wavelength from direct simulation are close to analytical calculation in lower elastic modulus, while it grows a little higher for the larger elasticity”.

Why the wrinkling wavelength from direct simulation are close to analytical calculation in lower elastic modulus and it grows a little higher for the larger elasticity? Explain briefly.

Q16. P5, L 9-12, “the direct simulation has been achieved with 1 imperfection”.

What happens to the direct simulation results if there are a greater number of imperfections? Do you need to change the boundary conditions if there are a greater number of imperfections?

Please justify your comparison with all techniques.

Q17. P5, L 22-23, “Fig. shows the critical strain as function of elasticity of thin film for different elasticity of substrate”.

Please indicate the Figure No. in the text.

Q18. P5, L 27-29, “Besides, the substrate elasticity has the same tendence as the thin film as the critical strain has been reduced at lower substrate elasticity”.

Please correct the sentence structure and English grammar.

Q19. P6, L 2-3, “Fig. shows wrinkling amplitude variation as function of applied strain”.

Please indicate the Figure No. in the text

Q20. P6, L 10. Fabrication of thin film wrinkling and material properties extraction

The manuscript lacks tables. Please display the wrinkling wavelengths of the thin films extracted on a table and compare it with the results of other techniques. It will be easy to understand.

Author Response

The answers for the reviewer's comments have been prepared in the attached file.

Reviewer 3 Report

Reviewer Comments

Comments to the author:

In this paper, analysis of ‘Direct Numerical Simulation of Surface Wrinkling for Thin Metal Film Material Properties Extraction was debated. The authors investigated the effect of the number of imperfections on wrinkling characteristics first and wrinkling wavelength depending on the elastic moduli of the associated materials has been then prepared for material properties extraction. Following comments should be considered.

1.      The abstract section should be improved with major finding and applications.

2.      In the abstract, at least two major findings should be included.

3.      What are the achievements of previous studies based on a numerical basis? Also, describe what has not been achieved?

4.      Research questions are needed. This would guide the author to structure logical analysis of results. Logical questions are expected. This would help readers to link what is known in the literature with the novelty of this study.

5.      The Introduction should make a compelling case for why the study is useful along with a clear statement of its novelty or originality by providing relevant information and providing answers to basic questions such as: 

a)      What is already known in the open literature?

b)      What is missing (i.e., research gaps)?

c)      What needs to be done, why and how?

d)     Clear statements of the novelty of the work should also appear briefly in the Abstract and Conclusions sections.

6.      The discussion seems inadequate, and this is too short for a reputable international journal.

7.      What software is used for the simulations? Was the code for the implemented by the authors or a function already existing in the software was used? If the code for the numerical method was taken from another publication or is part of the software used, please cite the resource.  

The conclusion must answer whether the proposed method can solve the research problem and achieve the objective. What is the most important result?

Author Response

The answers for the comments have been attached.

Round 2

Reviewer 1 Report

The authors have made the necessary changes, so I propose the manuscript for publication.